# Preparation of a Nano-Laminated Sc_2_SnC MAX Phase Coating on SiC Fibers via the Molten Salt Method

**DOI:** 10.3390/ma18112633

**Published:** 2025-06-04

**Authors:** Chenyang Wang, Lexiang Yin, Peng Li, Qing Huang

**Affiliations:** 1School of Materials Science and Chemical Engineering, Ningbo University, Ningbo 315211, China; wangchenyang@nimte.ac.cn; 2Zhejiang Key Laboratory of Data-Driven High-Safety Energy Materials and Applications, Ningbo Institute of Materials Technology and Engineering, Chinese Academy of Sciences, Ningbo 315201, China; yinlexiang@nimte.ac.cn

**Keywords:** MAX phase coating, molten salt synthesis, SiC fiber, pyrolytic carbon, CVD

## Abstract

The incorporation of MAX phase interface layers into silicon carbide (SiC) composites has been shown to significantly enhance mechanical properties, particularly under irradiation conditions. However, conventional Ti-based MAX phases suffer from thermal instability and tend to decompose at high temperatures. In this work, an Sc_2_SnC coating was successfully synthesized onto the surface of SiC fibers (SiC_f_) via an in situ reaction between metals and pyrolytic carbon (PyC) in a molten salt environment. The PyC layer, pre-deposited by chemical vapor deposition (CVD), served as both a carbon source and a structural template. Characterization by SEM, XRD, and Raman spectroscopy confirmed the formation of Sc_2_SnC coatings with a distinctive hexagonal flake-like morphology, accompanied by an internal ScC_x_ intermediate layer. By turning the Sc-to-Sn ratio in the molten salt, coatings with varied morphologies were achieved. ScC_x_ was identified as a critical intermediate phase in the synthesis process. The formation of numerous defects during the reaction enhanced element diffusion, resulting in preferential growth orientations and diverse grain structures in the Sc_2_SnC coating.

## 1. Introduction

Silicon carbide (SiC)-based ceramics and their composites (SiC_f_/SiC) are regarded as promising materials for components in light water reactors (LWRs) and advanced fission reactors, including high-temperature gas-cooled reactors (HTGRs), fluoride-salt-cooled high-temperature reactors (FHRs), and gas-cooled fast reactors (GFRs) [1]. The interfacial layer between SiC fibers (SiC_f_) and the SiC matrix plays a crucial role in integrating the mechanical, thermal, and electrical properties of the fibers and the matrix. This layer can significantly improve the composite’s strength, fracture toughness, resistance to radiation, oxidation, and corrosion, as well as thermal conductivity. However, under nuclear irradiation conditions, it is necessary to develop an interface layer that can withstand high temperatures and radiation, because conventional interfacial layers are prone to structural degradation and performance failure [2]. For example, pyrolytic carbon (PyC) undergoes shrinkage–swelling–amorphization structural evolution under neutron irradiation, leading to significant interface delamination. In hexagonal boron nitride (h-BN), the B element undergoes transmutation (producing He) during neutron exposure, resulting in interfacial layer damage. Similarly, multilayer interfaces composed of PyC and h-BN experience structural disruption under long-term irradiation, which reduces interfacial shear strength and increases frictional stress, making them suboptimal for interface layer applications [1,3].

In recent years, ternary layered MAX phase (a class of nano-laminated materials composed of an early transition metal (M), an A-group element (A), and C, N, B, and/or P (X) [4]) materials have attracted attention as candidates for nuclear-grade interfacial layer [5]. MAX phases have a hexagonal layered crystal structure similar to PyC and h-BN. Under applied stress, they exhibit deformation characteristics, such as slip, buckling, and kinking, which can deflect interfacial cracks and enhance the toughness of composite materials [6]. Additionally, MAX phase materials possess high thermal conductivity, corrosion resistance, and radiation resistance [7,8,9,10,11]. As interfacial materials, they contribute positively to both the thermal and radiation tolerance of the composites. Tallman et al. conducted a neutron transmutation analysis and found that the specific activity of MAX phases, such as Ti_3_SiC_2_, Ti_3_AlC_2_, and Ti_2_AlC, after 10, 30, and 60 years of exposure to thermal and fast neutron spectra is comparable to that of SiC and three orders of magnitude lower than that of nickel-based alloys, like Alloy 617 [7,11,12]. Despite their excellent radiation resistance, coating MAX phases onto SiC_f_ remains a technical challenge [13]. Filber-Demut et al. utilized the electrophoretic deposition method to coat the SCS-6 type of SiC_f_ with Ti_3_SiC_2_ powder and also demonstrated significant improvements in debonding strength and interfacial friction resistance in ceramic matrix composites [14]. However, electrophoretic deposition is limited to micron-sized Ti_3_SiC_2_ particles and often fails to achieve uniform coverage, posing a significant challenge for creating homogeneous interfacial layers.

Our working group was the first to developed a simple method for fabricating carbide/MAX phase interfacial layer via an in situ reaction in a molten salts bath [15]. We further utilized this technique to synthesize MAX phase interfacial layers, such as Ti_2_AlC and Ti_3_SiC_2_, on the surfaces of carbon fibers (C_f_), and composite materials were then fabricated using the polymer impregnation and pyrolysis process (PIP). After simulated ion irradiation, these composites exhibited good structural stability and had a good irradiation swelling behavior with SiC [16,17].

However, titanium-based MAX phases decompose into TiC at elevated temperatures (~1400 °C) and show poor oxidation resistance above 1200 °C [18]. This decomposition decreases the shear strength of the composite at high temperatures and necessitates precise temperature control during the subsequent SiC matrix processing. Therefore, it is necessary to develop a high-temperature-stable MAX phase interfacial layer.

According to previous reports, non-transition metal M_2_AX compounds containing Sc, Y, and Lu atoms at the M-site are considered to have a stable structure at high temperatures (~1400 °C) [19]. In addition, rare earth elements are known to impart unique properties [18,20,21], such as enhanced irradiation resistance, as observed in aluminum alloys [22] and likely in the in-plane-ordered RE-i-MAX phases [23]. Notably, Sc_2_SnC shows lower values of elastic constants (i.e., C_11_, C_33_, C_44_, and C_66_) compared to Ti_2_AlC, Ti_3_AlC_2_, and Ti_3_SiC_2_, indicating better deformability and processability.

In the present work, we first used the CVD method to coat a PyC layer onto the surface of SiC_f_. These carbon-coated fibers were then immersed in a molten salt environment to in situ synthesize a Sc_2_SnC MAX phase interfacial layer. The effects of the PyC layer structure, the sequence of element addition, element diffusion, and precursor composition on the quality of the resulting MAX phase interfacial layer were systematically investigated.

## 2. Experimental Details

### 2.1. Materials

Silicon carbide fiber (SiC_f_, Cansas-3303) was supplied by Fujian Liya New Material Co. Ltd. (Zhengzhou, China), and carbon fiber (C_f_, 3 k, T300) was provided by Toray Co., Ltd., Tokyo, Japan. Scandium powder (Sc, 300 mesh) and tin powder (Sn, 300 mesh) were sourced from Shanghai Pantian Powder Materials Co., Ltd. (Shanghai, China). Sodium chloride (NaCl, 99.5%, Aladdin, Shanghai, China) and potassium chloride (KCl, 99%, Aladdin, China) were used as the inert salt bath.

### 2.2. Preparation of the PyC Layer

SiC_f_ tow was first de-sized under vacuum at 600 °C over 2 h. A series of PyC interphase layers were then deposited using chemical vapor deposition (CVD). The fiber tow was placed inside a graphite sleeve lined with graphite paper for demolding. The assembly was heated in a quartz tube (Φ80 × 2 mm × 1.8 m) to a final pyrolysis temperature of 900–1100 °C at a ramp rate of 5 °C/min using a furnace (TL1200-1200, Boyuntong, Nanjing, China). After reaching the target temperature, the system was allowed to cool naturally to room temperature under vacuum. The base vacuum inside the furnace tube was maintained below 1 Pa before deposition. During the deposition, stable airflow was introduced into the graphite sleeve. Methane (CH_4_) and acetylene (C_2_H_2_) were used as carbon sources, while argon (Ar) served as both a dilution and carrier gas. The structure of the PyC interphase layer was controlled by adjusting CVD parameters, including temperature, residence time, and flow-field-related factors. The flow field is regulated using an automated gas supply system, controlling pressure, gas flux, and precursor composition. The PyC content in the in situ reaction was estimated based on its layer thickness. The final sample was designated as SiC_f_/PyC.

### 2.3. In Situ Synthesis of Sc_2_SnC Coating

The Sc_2_SnC coating was synthesized through the reaction between carbon, scandium powder, and tin powder in a molten salt environment. NaCl-KCl eutectic salt (melting point of ~660 °C) was used as the inert salt bath. The powders were mixed in a stoichiometric ratio of Sc:Sn = 2:1.1 (mol), unless otherwise specified. The tin content was increased to compensate for possible weight loss at high temperatures, as tin has a relatively low melting point, similar to the preparation of Sc_2_SnC MAX phase. The starting powders of Sc and Sn were combined with the NaCl-KCl salt in a molar ratio of (Sc + Sn):(NaCl-KCl) = 1:20 and thoroughly ground in an agate mortar. The as-prepared SiC_f_/PyC or C_f_ was placed in the alumina crucible and covered with the mixed powder. The crucible was then placed in a tubular furnace (SGL-1700, SIOMM, Shanghai, China) and heated at a rate of 5 °C/min under argon atmosphere. A range of temperatures and dwell times were tested to evaluate their effects on the composition and thickness of the interface layer. After the reaction, the sample was thoroughly washed, filtered, and dried at 40 °C under vacuum to remove the inert salt.

### 2.4. Characterization

The microstructure of the samples was examined by a field-emission scanning electron microscope (SEM, HITACHI Regulus 8230, Tokyo, Japan) equipped with an energy-dispersive spectroscopy (EDS, Detector6, Bruker XFlash, Berlin, Germany) system. X-ray diffraction (XRD) analysis was performed using an X-ray diffractometer (Bruker ADVANCE D8, Berlin, Germany) using Cu-K_α_ radiation at a scan rate of 0.02°/s. Raman spectra were obtained using a confocal Raman spectrometer (HORIBA LabRAM HR Evolution, Tokyo, Japan) with a 532 nm excitation wavelength.

## 3. Results

### 3.1. PyC Pre-Film Synthesis and Characterization

The microstructure of the as-produced PyC influences the structure of Sc_2_SnC prepared by the molten salt method. To obtain a uniform and smooth carbon layer, we investigated the structural variations of PyC on SiC_f_ under different preparation conditions. The temperature range (900 °C, 1000 °C, 1100 °C) was selected based on a comprehensive consideration of the pyrolysis temperature range of acetylene and methane, as well as the impact of the temperature on SiC_f_. Temperatures exceeding 1200 °C can damage the carbonized fibers [24,25,26,27,28]. Figure 1 shows the morphology and Raman spectra of PyC prepared at different temperatures. The surface morphology reveals that the PyC coating uniformly conforms to SiC_f_, forming an intact, dense, and smooth layer (Figure 1(a-1–c-1)). Upon magnification, some abnormally large particles or granules are visible (indicated by white arrows in Figure 1(b-2,c-2)), which may be related to the rapid pyrolysis of acetylene, as discussed in Figure 2. Additionally, from the cross-sectional morphology, it is evident that as the temperature increases, the PyC layer becomes denser, and the laminar texture becomes more pronounced. Within the temperature range of 900~1100 °C, no noticeable gaps are observed at the interface between the PyC and SiC_f_, indicating a strong interfacial bond between the coating and the fibers. The Raman spectra at different temperatures are shown in the Figure 1(a-4–c-4). Due to the disordered nature of carbon materials, PyC tend to exhibit similar Raman spectra, despite having considerable structural differences [29,30,31]. To investigate the influence of temperature, spectral deconvolution using five Raman bands—D_1_, G, D_2_, D_3_ and D_4_—was performed to qualitatively assess the defect levels in the PyC layer through the I_D_/I_G_ ratio [32,33]. Higher temperature results in a lower I_D_/I_G_ ratio (from 3.86 to 2.89, compared to 3.70 for de-sized C_f_ T300), indicating that the PyC coating is more compact and has fewer defects. Figure 2 shows the microstructural morphology of the PyC coatings prepared with different ratios of CH_4_ and C_2_H_2_. It can be observed that as the C_2_H_2_ concentration increases, larger particles appear on the surface. This may be due to the higher carbon deposition rate of C_2_H_2_, which results in insufficient time for carbon atoms to diffuse, leading to local aggregation. From the cross-sectional morphology, it can be seen that when the CH_4_-C_2_H_2_ ratio is 90:60, the resulting PyC coating has a high density and a strong bond with the SiC_f_. Additionally, from the Raman spectra, its I_D_/I_G_ ratio is relatively lower, indicating that the PyC has a more compact structure (Figure 2b). Given the strict requirements for interface layer thickness, an overly thick layer may lead to interface delamination or stress concentration, while an excessively thin layer may fail to prevent crack propagation. Therefore, precise control of the PyC thickness is crucial. Figure 3 shows the relationship between the PyC thickness (*T*) and time (*t*) at 1000 °C with a CH_4_-C_2_H_2_ ratio of 90:60. The statistical data is fitted to a logistic function (1), and it is clear that PyC thickness and deposition time show a nonlinear relationship (Table 1). This could be attributed to the changes in the adsorption and pyrolysis rates of methane and acetylene gases over time.(1)$$\begin{array}{c} T=513.14-\frac{522.10}{1+\frac{t^{1.95}}{2069.50}} \end{array}$$

### 3.2. MAX Phase Coating

#### 3.2.1. Carbon Fiber Coating (C_f_/ScC_x_/Sc_2_SnC)

The reaction kinetics in the molten salt synthesis of MAX phases vary depending on the carbon structure. To explore an optimal preparation process, we used C_f_ as a substitute for PyC coatings. This choice was based on the similarity in the amorphous structures and defect levels of C_f_ and PyC. Furthermore, C_f_ is more readily available and allows for better control of single variables. Building on our previous work on synthesizing carbon-containing MAX phases at low temperatures using the molten salt method [34,35,36,37], we adopted two different synthesis routes for MAX phase coatings using C_f_, as follows:Direct Molten Salt Route: Sc, Sn, and C_f_ were directly mixed to form Sc_2_SnC.Two-Step Molten Salt Route: Sc and C_f_ were first mixed to form an intermediate phase, ScC, followed by the introduction of Sn to synthesize Sc_2_SnC.

In both routes, the reactions were conducted at 1000 °C for 3 h, with a Sc-to-C_f_ molar ratio of 1:1. The XRD patterns of Sc_2_SnC synthesized via these two routes are shown in Figure 4. It is evident that neither approach resulted in the significant formation of Sc_2_SnC. In the direct molten salt route (Route 1), the primary phase formed was ScC ($Fm\bar{3}m$, PDF#97-004-3524), with only trace amounts of Sc_2_SnC (P6_3_/mmc, [19]). In the two-step route (Route 2), the intermediate phase ScC was successfully synthesized first; however, after introducing Sn, Sc_2_SnC was still not observed. Instead, small amounts of Sc_2_O_3_ and Sc_2_OC were detected, possibly due to oxidation interference. Raman spectroscopy analysis, as shown in Figure 4b, further supports these findings. In Route 1, three characteristic peaks were observed at 139, 207, and 217 cm^−1^, corresponding to the ω_1_, ω_2_, and ω_3_ vibrational modes of the 211-type MAX phase. In contrast, Route 2 exhibited only the ω_3_ peak, with no other distinct features [38,39,40,41]. Notably, the ω_4_ vibrational mode at approximately 380 cm^−1^ was not clearly detected. Additionally, peaks observed in the 380–800 cm^−1^ range correspond to carbide materials [42,43], suggesting that the broadening of the ScC peaks in the XRD spectrum may be attributed to non-stoichiometric scandium carbide or ScC_x_ with a high defect density.

Figure 5 further presents the SEM morphology of the Sc_2_SnC coatings synthesized via the two routes. In Route 1, a uniform ultra-thin coating composed of nanocrystals was formed. In Route 2-1, ScC_x_ exhibited a well-defined coating morphology on the fiber surface, accompanied by a small number of particles. Following the introduction of Sc and Sn, a significant number of nanocrystals appeared on the surface, identified as granular Sc_2_O_3_ and nano-platelet-like Sc_2_SnC. Additionally, partial oxidation led to the formation of accordion-like Sc_2_OC or complete oxidation into Sc_2_O_3_, causing the ScC_x_ layer to peel off, as indicated by the white arrows in Figure 5(c-1). Consequently, in Route 2-2, oxidation resulted in the loss of Sc, which restricted the reaction process. Given the low conversion rate of Sc_2_SnC, it is likely that element diffusion and reaction kinetics are the key controlling factors, which will be further discussed. Based on this analysis, the direct synthesis method (Route 1) is more favorable for the fabrication of Sc_2_SnC coatings.

As the Sc_2_SnC synthesized by the direct method shows only faint diffraction peaks in Figure 4a, the reaction appears incomplete under a relatively stoichiometric ratio, likely due to insufficient diffusion time. To address this, we extended the reaction duration. As shown in Figure 6, prolonging the reaction time significantly increased the content of both Sc_2_SnC and ScC_x_, with peak concentrations reached after 4 h of reaction. However, with further extension, partial transformation of ScC_x_ occurred due to its metastable nature and tendency to oxidize. Specifically, ScC_x_ either oxidized to form Sc_2_OC ($\mathrm{Fm}\bar{3}m$, PDF#97-015-6683) or was reduced to Sc_3_C_4_ (P4/mnc, PDF#97-007-1145). As the fibers were held at a high temperature for a longer period, the transformation to Sc_2_OC continued, while the formation of Sc_3_C_4_ was limited due to the restricted diffusion of carbon. Both transformation pathways are accompanied by lattice expansion, which can induce stress or even cause cracking. Overall, excessive reaction time can damage the fibers, leading to coating delamination or pulverization.

Figure 7 presents the microstructural morphology of Sc_2_SnC coatings formed at different reaction durations. At 3 h, the Sc_2_SnC appears as isolated hexagonal sheets that seem to be “embedded” within C_f_. This unique morphology resembles the loading of nanocrystals rather than a uniform coating, which is confirmed by mapping in Figure 8. As the reaction proceeds, the Sc_2_SnC sheets gradually accumulate and form a dense coating through a radial growth mode. After 4 h of treatment (as shown in Figure 7b), the material exhibits a well-defined crystalline structure. Interestingly, as the reaction continues, these distinctive crystals grow independently, reaching a maximum size of approximately 5 µm. Correspondingly, the thickness of the Sc_2_SnC coating increases to around 50 nm, 500 nm, and 2.9 µm after 3, 4, and 5 h of reaction, respectively. A scandium carbide (ScC_x_) layer is observed only in the 5 h samples. It forms a dense interfacial layer between the Sc_2_SnC coating and the C_f_ substrate, and its thickness increases along with the Sc_2_SnC layer. Although it is challenging to distinguish ScC_x_ from Sc_2_OC and Sc_3_C_4_ using SEM imaging and elemental mapping, these by-products possess larger lattice parameters and introduce residual stress within the coating. The stress may ultimately lead to the formation of cracks, a phenomenon that already observed in the 4 h sample.

According to the growth mechanism of molten salt synthesis for MAX phases, the general reaction Equations (2)–(4) are listed in order of increasing temperature. Eutectic salts create an ionized environment at relatively low temperatures, facilitating the formation of Sc-Sn intermetallic compounds and ScC_x_. ScC_x_ is considered the most critical intermediate product in the reaction. Due to the high melting points of both scandium and carbon, the carbide layer can only form at around 1000 °C, primarily at reaction interfaces characterized by abundant defects and preferred crystallographic orientations. Once ScC_x_ is formed, the reaction between the Sc-Sn intermetallic compound and scandium carbide can proceed. With the assistance of molten salts, elemental species are rapidly transported to the reaction interface. Meanwhile, the abundant defects in the non-stoichiometric ScC_x_ provide pathways for the diffusion of Sc and Sn. As a result, Sc_2_SnC exhibits a high nucleation and growth rate, which explains the absence of a distinct ScC_x_ layer during the early stages of the reaction. Furthermore, the presence of multiple close-packed crystal planes and a high c/a ratio leads to anisotropic growth behavior in Sc_2_SnC, typically resulting in flake-like structures.

(2)
Sc + Sn ↔ Sc-Sn (intermetallic)



(3)
$$\mathrm{Sc}+C\overset{molten\;chloride}{\to }\;{\mathrm{ScC}}_{x}$$



(4)
$${\mathrm{ScC}}_{x}+\mathrm{Sc}-\mathrm{Sn}\;\overset{molten\;chloride}{\to }\;{\mathrm{Sc}}_{2}\mathrm{SnC}$$


The reaction follows a template synthesis mechanism in which Sc_2_SnC inherits the structure of its precursors—from the graphite-like PyC layer to the cubic ScC phase. Specifically, under suitable conditions, Sc_2_SnC nucleates and grows along the [111] plane or its equivalent planes, such as [$1\bar{11}$], which is rotated 75.53° from the [111] plane of cubic crystal. This growth is typically aligned parallel or nearly perpendicular to the fiber axis. In addition, the slight preferred orientation of ScC in the {200} plane provides numerous nucleation sites along the {111} family of planes. As a result, Sc_2_SnC flakes initially grow nearly perpendicular to the surface of the C_f_. Once a continuous and dense Sc_2_SnC coating is formed, the diffusion channels are largely closed. At this point, the metastable ScC phase (with a formation enthalpy of ΔH_form_ = −0.138 eV, according to the Materials Project) may transform into the thermodynamically stable Sc_3_C_4_ phase (ΔH_form_ = −0.408 eV) via carbon diffusion. This transformation could lead to structural pulverization and delamination. Ultimately, a multilayered interface consisting of ScC_x_ and Sc_2_SnC is formed on the surface of C_f_.

#### 3.2.2. Silicon Carbide Fiber Coating (SiC_f_/ScC_x_/Sc_2_SnC)

Based on the growth mechanism observed on C_f_, the reaction was transferred onto SiC_f_ substrates coated with an approximately 370 nm PyC layer. The synthesis was performed at 1000 °C for 3 h, using varying Sc:Sn molar ratios. Figure 9a presents the XRD patterns of all SiC_f_ coatings. As the reaction progresses, the amorphous peak gradually diminishes, while the characteristic diffraction peaks of Sc_2_SnC and ScC_x_/Sc_2_OC start to appear. Notably, the Sc_3_C_4_ phase is only observed in the sample with a 3:2 ratio (SiC_f_/Sc_2_SnC-3:2). The composition of the interface is further confirmed by the Raman spectra shown in Figure 9b. Four distinct peaks at 139, 207, 217, and 380 cm^−1^ correspond to the typical vibrational modes of the 211-type MAX phase. Among these, the peaks at 207 and 380 cm^−1^ show significant enhancement, indicating structural differences in the coating, even though the XRD patterns remain relatively unchanged. Additionally, a previously unreported peak appears around 320 cm^−1^, for which no known vibrational mode has been identified. Some studies have observed similar satellite peaks in 211 MAX phases; although, they have not been extensively studied. As expected, Sc_2_SnC coatings formed under different Sc:Sn ratios exhibit three distinct morphologies. Under the stoichiometric condition (Sc:Sn = 2:1), Sc_2_SnC displays the same flake-like morphology as seen on C_f_ substrates, as shown in Figure 10a. The elemental mapping of the scaly surface is presented in Figure 11h–l. The cross-sectional image reveals that a significant amount of PyC remains unreacted.

When the Sc:Sn atomic ratio is increased to 3:1, the resulting coating becomes dense and continuous, consisting of nanocrystalline structures. Notably, some surface particles exhibit abnormal grain growth, forming equiaxed crystals with diameters up to 1.8 μm. However, this rapid grain growth leads to structural inhomogeneity and localized pulverization. In cross-sectional observations, the Sc_2_SnC layer shows weak adhesion to the unreacted PyC layer, likely due to residual stresses arising from mismatched thermal expansion coefficients.

At a Sc:Sn ratio of 3:2, a more uniform and finer Sc_2_SnC coating is formed, with a thickness of approximately 1.4 μm. No delamination is observed between the coating and the SiC_f_ substrate. The measured thickness is consistent with the theoretical lattice expansion from PyC (a = 0.247 nm, c = 0.693 nm; graphite, PDF#97-061-7290) to Sc_2_SnC (a = 0.337 nm, c = 1.464 nm, Ref. [19]), indicating the complete transformation of PyC. This coating exhibits excellent uniformity and strong interfacial adhesion, with surface cracks attributed only to the formation of by-products.

Cross-sectional elemental mapping (Figure 11a–g) confirms the uniform distribution of scandium and tin within the coating, as well as the structural integrity of the SiC_f_ substrate. In addition, the high oxygen concentration detected on the surface is likely due to the formation of Sc_2_OC and the adsorption of atmospheric oxygen.

## 4. Conclusions

(1)A dense and uniformly coated PyC layer was successfully deposited on SiC_f_ via CVD by precisely controlling the reaction temperature at 1000 °C and setting the CH_4_:C_2_H_2_ gas ratio to 90:60.(2)ScC_x_/Sc_2_SnC composite coatings were synthesized on the surfaces of C_f_ and SiC_f_ using a molten salt method. In the early stages of the reaction, isolated hexagonal Sc_2_SnC flakes nucleated and grew on the C_f_ surface. With the extended reaction time, these nearly vertically oriented flakes gradually accumulated to form a continuous coating, with the thickness progressively increasing from 50 nm to 500 nm and ultimately to 2.9 μm.(3)During the reaction process, the formation of ScC_x_ exhibited sluggish kinetics, making it a key intermediate that governed the overall reaction pathway. The high defect density and preferential orientation observed in ScC_x_ contributed to the distinctive microstructure and growth direction of the resulting Sc_2_SnC phase. Raman spectroscopy confirmed the presence of both ScC_x_ and Sc_2_SnC. However, due to its metastable nature, ScC_x_ is prone to phase transformation into Sc_2_OC and Sc_3_C_4_, which may lead to cracking, pulverization, interfacial debonding, and the eventual delamination of the coating—a challenge that remains difficult to mitigate.

## Figures and Tables

**Figure 1 materials-18-02633-f001:**
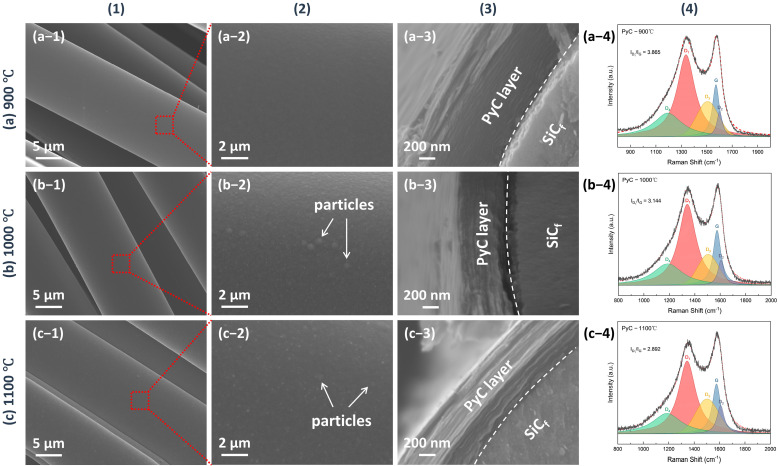
The morphology of PyC pre-film produced at (**a**) 900 °C, (**b**) 1000 °C, and (**c**) 1100 °C, for 60 min under 50 kPa in a CH_4_-C_2_H_2_-Ar atmosphere (CH_4_:C_2_H_2_:Ar = 90:60:200 sccm). (**1**) and (**2**) show the surface morphology of the PyC film, with (**2**) being a magnified view of a selected area in (**1**), (**3**) displays the cross-sectional morphology, and (**4**) presents the corresponding Raman spectra.

**Figure 2 materials-18-02633-f002:**
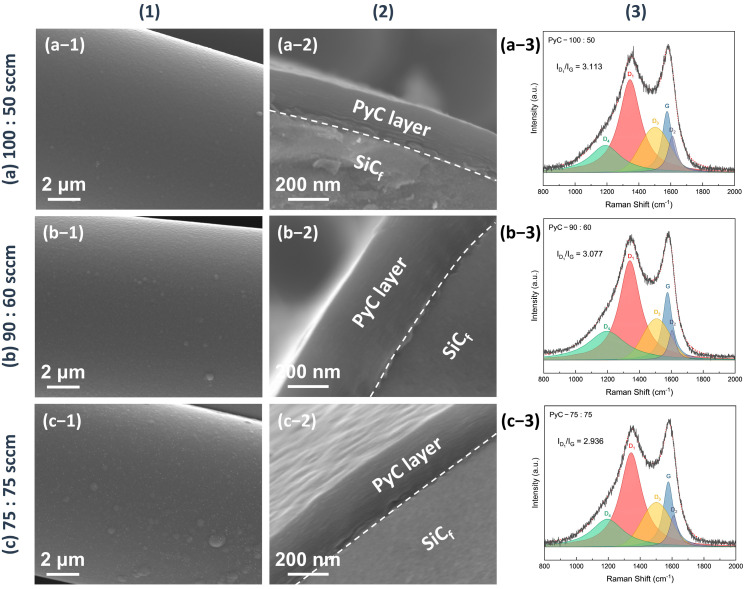
The morphology and Raman spectra of the PyC pre-film produced under different CH_4_ and C_2_H_2_ gas flow ratio—(**a**) 100:50 sccm, (**b**) 90:60 sccm, (**c**) 75:75 sccm—at 1000 °C and 50 kPa. (**1**) shows the surface morphology, (**2**) the cross-sectional morphology, and (**3**) the corresponding Raman spectra. The flow rate of the diluting gas (Ar) is fixed at 200 sccm.

**Figure 3 materials-18-02633-f003:**
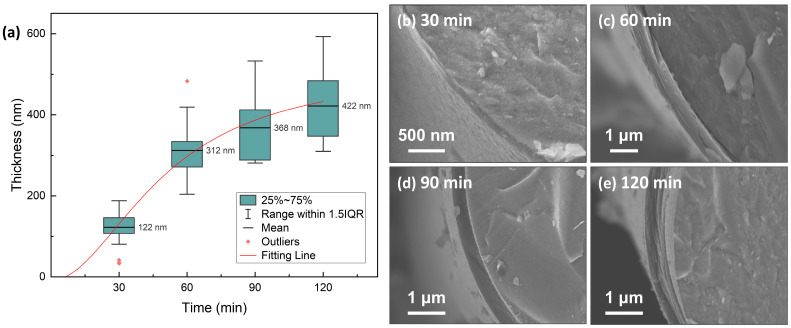
(**a**) Relationship between time and thickness of PyC pre-film. Typical samples deposited for (**b**) 30 min, (**c**) 60 min, (**d**) 90 min, and (**e**) 120 min are shown on the right.

**Figure 4 materials-18-02633-f004:**
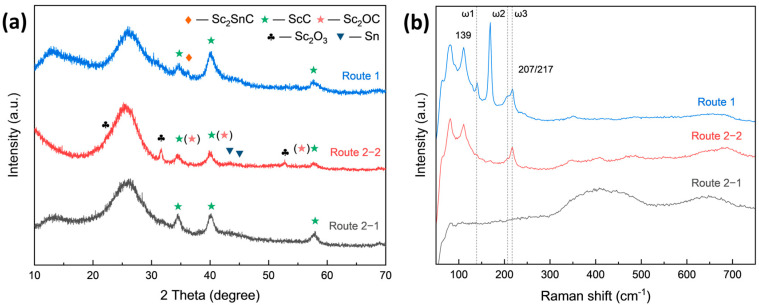
(**a**) XRD pattern and (**b**) Raman spectra of C_f_ reacted in Route 1 and Route 2.

**Figure 5 materials-18-02633-f005:**
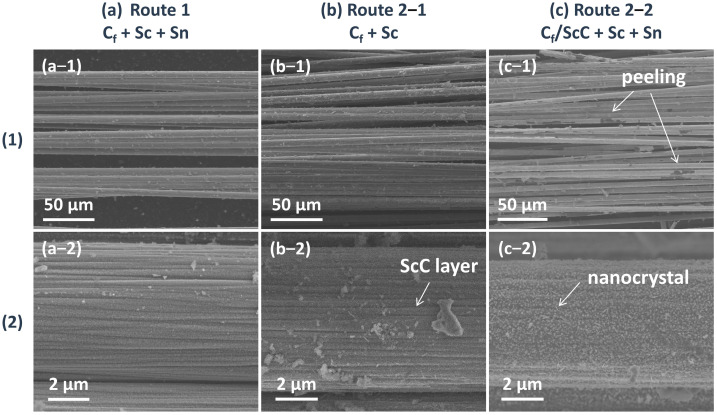
The (**1**) overview and (**2**) detailed surface morphology of C_f_ after (**a**) Route 1, (**b**) first step of Route 2, (**c**) second step of Route 2.

**Figure 6 materials-18-02633-f006:**
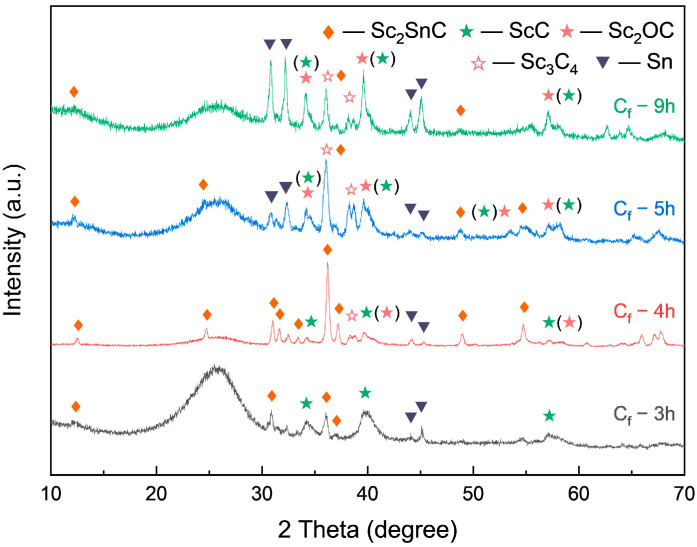
XRD pattern of C_f_ after long term dwelling in molten salt.

**Figure 7 materials-18-02633-f007:**
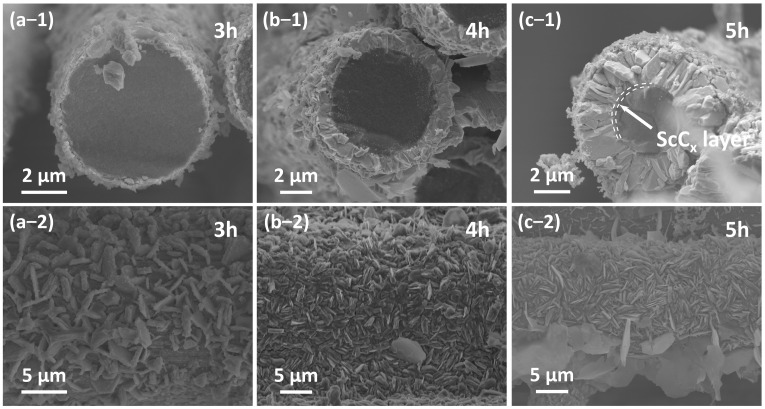
The cross-sectional and surface morphology evolution of C_f_ after (**a**) 3 h, (**b**) 4 h, and (**c**) 5 h reaction.

**Figure 8 materials-18-02633-f008:**
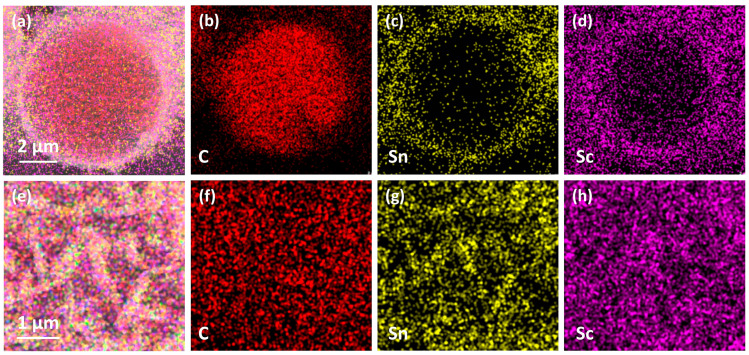
The mapping results of (**a**–**d**) cross section, (**e**–**h**) surface of C_f_—3h sample.

**Figure 9 materials-18-02633-f009:**
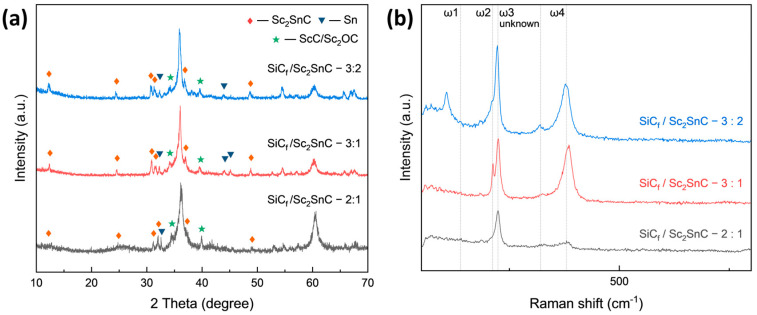
(**a**) XRD pattern and (**b**) Raman spectra of SiC_f_ with the PyC layer reacted within different ingredient ratios; the samples are named after SiC_f_/Sc_2_SnC—(Sc:Sn ratio).

**Figure 10 materials-18-02633-f010:**
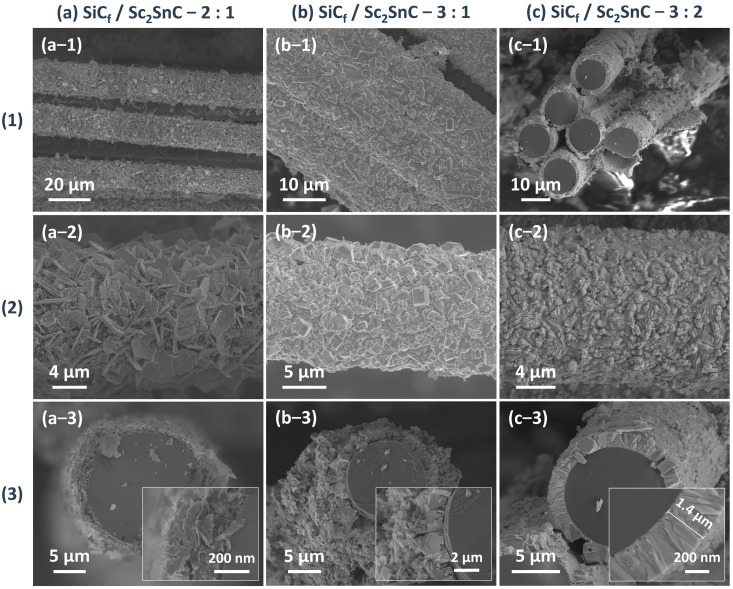
The morphology of the Sc_2_SnC coating produced within different ingredient ratios at 1000 °C for 3 h. (**1**) Overview, (**2**) surface, and (**3**) cross-section images. (**a**) SiC_f_/Sc_2_SnC—2:1, (**b**) SiC_f_/Sc_2_SnC—3:1, and (**c**) SiC_f_/Sc_2_SnC—3:2.

**Figure 11 materials-18-02633-f011:**
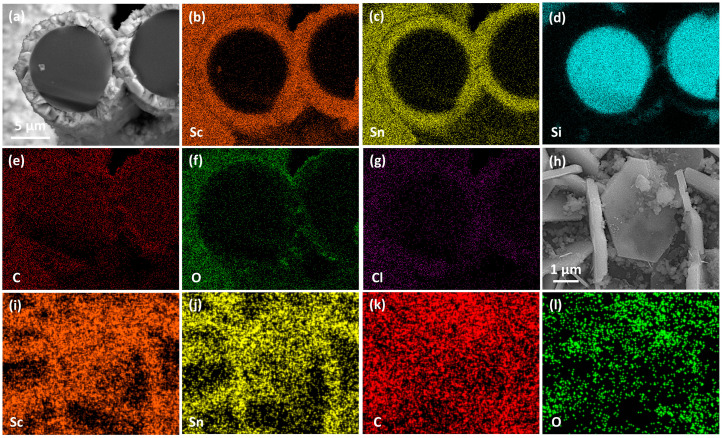
Elemental mapping of Sc_2_SnC coatings on SiC_f_. (**a**–**g**) Cross-sectional elemental distribution of the SiC_f_/Sc_2_SnC—3:2. (**h**–**l**) Surface elemental distribution of the SiC_f_/Sc_2_SnC—2:1.

**Table 1 materials-18-02633-t001:** Fitting details of the logistic function for the data in Figure 3a.

Argument	Value
Equation	$T=A_{2}+\frac{A_{1}-A_{2}}{1+{\left(\frac{t}{t_{0}}\right)}^{P}}$
$A_{1}$	−8.957 ± 76.068
$A_{2}$	513.14 ± 77.330
$t_{0}$	50.17 ± 9.965
P	1.954 ± 0.656
R-Square	0.73704

## Data Availability

The original contributions presented in this study are included in the article. Further inquiries can be directed to the corresponding author(s).

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
