# Peer review of "Preparation of a Nano-Laminated Sc_2_SnC MAX Phase Coating on SiC Fibers via the Molten Salt Method"

_materials, 2025, doi:10.3390/ma18112633_

Round 1

Reviewer 1 Report

Comments and Suggestions for Authors

The paper “Preparation of a Nano-laminated Sc2SnC MAX Phase Coating on SiC Fibers via the Molten Salt Method” presents a study on a molten-salt method of SiC fibers coating with a MAX phase. The paper has a well reasoned results, a good literature list. However, I miss some important things in Methods section, as well as the results presentations form.

  1. As long as the main idea of such a phase producing is to increase radiation resistance of the whole construction, I find it appropriate to mention in the very beginning that such a composite material is applied/ is perspective at application in conditions of strong radiation : condition 1, condition 2, etc….
  2. All abbreviations are supposed to be deciphered at a first mention, and later only abbreviations should be used.
  3. In the methods section I miss description of the equipment used during the phases preparation, such as furnaces, vacuum chambers, etc. The equipment should be presented in the form MODEL (Manufacturer, Country, City). Also, bring all equipment description to this form.

  1. Figures.

Fig. 1: incorrect sections designation. It should be (a) -900C, (b) -1000C, (c) – 1100C on the left and 1, 2, 3, 4 or I, II, III, IV on the top. There should not be a number “0” in the sections. Then it will be in the figure caption: 2) – surface, 3) – cross section, etc.

Fig. 2: the same.

Fig. 3: table should be given separately. Pictures should be bigger. Perhaps, as a different picture.

Fig 5: as in Figs. 1 and 2. On the left it should be 1) 2) and on the top it should be a) and b) / Then the figure caption should denote that a) means Route 1… ; b) means Route 2-1…. Etc.

Fig. 7. Should probably be separated into two or even more bigger figures.

Fig. 9 – the same as 1 and 2.  

  1. List of references

I recommend adding DOI to all cited references where applicable.

In this journal it is typical that up to first 10 authors of a work are mentioned. Only if there are more than 10 authors, one can write “Author 1 A.A., … , Author 10 A.A., etc.”

Author Response

Dear Reviewer: 

Thank you for your comments on improving the quality of our manuscript. We have made the necessary modifications based on your feedback, which have been marked in the “Modified Version for Comparison - Materials-3610652”. (Complete version for reading is also provided.) The details of the changes are as follows:

Reviewer #1: The paper “Preparation of a Nano-laminated Sc2SnC MAX Phase Coating on SiC Fibers via the Molten Salt Method” presents a study on a molten-salt method of SiC fibers coating with a MAX phase. The paper has a well-reasoned result, a good literature list. However, I miss some important things in Methods section, as well as the results presentations form.

1). As long as the main idea of such a phase producing is to increase radiation resistance of the whole construction, I find it appropriate to mention in the very beginning that such a composite material is applied/ is perspective at application in conditions of strong radiation: condition 1, condition 2, etc…

Response: Thanks very much for your professional guidance and suggestions. In accordance with your suggestion, we have elaborated on the potential applications of SiCf/SiC composites as structural materials in various types of nuclear reactors (see Introduction, lines 25–28). For example, we have cited relevant literature regarding the use of these composites in fuel cladding and channel boxes for light water reactors (LWRs), as well as in fuel assemblies and internal components for high-temperature gas-cooled reactors (HTGRs), fluoride-salt-cooled high-temperature reactors (FHRs), and gas-cooled fast reactors (GFRs), which are representative of advanced fission reactor systems.

In addition, we have cited relevant literature to support the neutron irradiation resistance of MAX phase materials when used as interfacial layers in composites (see Introduction, lines 50-56). We also included reports on the irradiation tolerance of scandium (Sc), such as its ability to significantly enhance the strength and irradiation resistance of aluminum alloys. Furthermore, we referenced studies showing that the incorporation of rare earth elements into MAX phases to form in-plane ordered RE-i-MAX phases might bring excellent irradiation resistance to the materials. (see Introduction, lines 76-77).

Changes at Introduction, lines 25-28, 50-56, 76-77, highlighted in blue for clarity.

 2). All abbreviations are supposed to be deciphered at a first mention, and later only abbreviations should be used.

Response: We appreciate your suggestion to enhance the clarity of the manuscript. All abbreviations are explained when they first appear.

Changes are marked as lines 29, 43-45, 57, 65-66, 80-81, 91, 95, 103-104, 138, 141, 144, 157, 159-161, 164, 194, 253, 262, 297, 303, 305, highlighted in blue.

3). In the methods section I miss description of the equipment used during the phases preparation, such as furnaces, vacuum chambers, etc. The equipment should be presented in the form MODEL (Manufacturer, Country, City). Also, bring all equipment description to this form.

Response: Details of the relevant equipment, including models and manufacturers, have been included in the Experimental section.

Changes locate in Experiment, lines 91, 98, 120, 127-131, highlighted in blue for clarity.

4). Figures.

Fig. 1: incorrect sections designation. It should be (a) -900C, (b) -1000C, (c) – 1100C on the left and 1, 2, 3, 4 or I, II, III, IV on the top. There should not be a number “0” in the sections. Then it will be in the figure caption: 2) – surface, 3) – cross section, etc.

Fig. 2: the same.

Fig. 3: table should be given separately. Pictures should be bigger. Perhaps, as a different picture.

Fig 5: as in Figs. 1 and 2. On the left it should be 1) 2) and on the top it should be a) and b) / Then the figure caption should denote that a) means Route 1…; b) means Route 2-1…. etc.

Fig. 7. Should probably be separated into two or even more bigger figures.

Fig. 9 – the same as 1 and 2.

Response: Thank you for your valuable suggestions. All the figures and tables in the manuscript have been revised according to your suggestions, which have indeed made the presentation clearer and more understandable.

Changes in Figure 1-3, 5, 7-8, 10, and Table 1, and corresponding numbering and/or figure caption, highlighted in blue.

5). List of references

I recommend adding DOI to all cited references where applicable.

In this journal it is typical that up to first 10 authors of a work are mentioned. Only if there are more than 10 authors, one can write “Author 1 A.A., …, Author 10 A.A., etc.”

Response: Thank you for your valuable suggestions. The formatting of all references, including the addition of DOI where applicable, has been revised accordingly.

Changes in References, lines 376-491, highlighted in blue.

Reviewer 2 Report

Comments and Suggestions for Authors

This paper presents a technically sound and innovative approach to coating design using Sc-based phases on ceramic fibers. It’s particularly strong in synthesis methodology, microstructural evolution, and phase analysis. Alaways is a space for improwment.

In order  to improve the clarity and flow I recommend:

  1. Shorten long sentences (many are 3-4 clauses long).
  2. Explicitly state objectives earlier (your "present work" appears rather deep into the text).
  3. Highlight why Sc₂SnC MAX phase is significant compared to Ti-based MAX clearly upfront.

Below are the sentences that need to be improved:

Line 29: „instance” change  → example

Line 35; „fewer ideal candidates”  → less ideal candidates

Line: 89:

“After reaching the target temperature, the system was naturally cooled...” → allowed to cool naturally

Line 109: “varying temperatures and dwelling time” → “a range of temperatures and dwell times”

Line 111: “adequately washed” → “thoroughly washed”

Line 115: “electronic microscope” → “electron microscope”

Line 118: “Raman spectroscopy” → “Raman spectrometer”

Line 128: It was observed that when the temperature exceeds 1200 ℃, damage to the carbonized fibers occurs..." →"Temperatures above 1200 °C cause damage to the carbonized fibers..."

Line 141: "Deconvolution arithmetic" → "Spectral deconvolution using five Raman bands..."

Line 159: "Time and thickness do not belong to a linear relationship" → "PyC thickness and deposition time show a nonlinear relationship..."

Comments on the Quality of English Language

english fine

Author Response

Dear Reviewer: 

Thank you for your comments on improving the quality of our manuscript. We have made the necessary modifications based on your feedback, which have been marked in the “Modified Version for Comparison - Materials-3610652”. (Complete version for reading is also provided.) The details of the changes are as follows:

Reviewer #2: This paper presents a technically sound and innovative approach to coating design using Sc-based phases on ceramic fibers. It’s particularly strong in synthesis methodology, microstructural evolution, and phase analysis. Always is a space for improvement. In order to improve the clarity and flow I recommend:

1). Shorten long sentences (many are 3-4 clauses long).

Response: Thank you for your valuable suggestions. We agree that long sentences may obscure the clarity of expression. Therefore, we have shortened lengthy sentences and polished the language throughout the manuscript to improve clarity. All the language changes are listed in 4).

2). Explicitly state objectives earlier (your "present work" appears rather deep into the text).

Response: Thank you for your valuable comment. We have moved the statement of the research objectives earlier in the introduction section to make them more explicit and easier for readers to identify. As noted in lines 32–35 of the introduction, we explained the significance of preparing a suitable interface layer. The abstract also clearly states the successful fabrication of the Sc2SnC coating. However, due to the weak adhesion between the Sc2SnC coating and the SiC fibers, the coating tends to peel off easily. This limitation has prevented us from further testing its irradiation resistance and high-temperature stability. Therefore, substantial further work is still needed to achieve our research goals.

Refine in Introduction, lines 32–35, marked in blue.

3). Highlight why Sc₂SnC MAX phase is significant compared to Ti-based MAX clearly upfront.

Response: We appreciate your insightful question. The issue of Ti-based MAX phases decomposing into TiC at elevated temperatures (~1400 °C), resulting in poor thermal stability, has been well documented in the literature. As a result, we selected the Sc2SnC MAX phase as an interface layer material based on our previous density functional theory (DFT) calculations, which demonstrated that Sc2SnC exhibits superior thermodynamic stability. Moreover, existing studies have shown that the incorporation of rare-earth elements into MAX phases can significantly enhance their radiation resistance and high-temperature stability. The experimental objective of this work was to pre-synthesize a uniform and well-adhered Sc2SnC interface layer on SiCf fibers, providing a robust foundation for subsequent evaluation of its irradiation and thermal stability. However, during the synthesis process, the formation of an intermediate ScCx phase may have triggered phase transformations, which in turn led to partial delamination of the coating. This delamination has limited our ability to comprehensively assess the high-temperature and radiation resistance of the interface layer. Several underlying mechanisms remain unclear, and further systematic studies are required to better understand and optimize the structural stability of the Sc2SnC coating.

Changes at Introduction, lines 70-79, highlighted in blue.

4). Below are the sentences that need to be improved

Line 29: “instance” → “example”

Line 35; “fewer ideal candidates”→ “less ideal candidates”

Line: 89: “After reaching the target temperature, the system was naturally cooled...” → “allowed to cool naturally”

Line 109: “varying temperatures and dwelling time” → “a range of temperatures and dwell times”

Line 111: “adequately washed” → “thoroughly washed”

Line 115: “electronic microscope” → “electron microscope”

Line 118: “Raman spectroscopy” → “Raman spectrometer”

Line 128: “It was observed that when the temperature exceeds 1200 ℃, damage to the carbonized fibers occurs...” → “Temperatures above 1200 °C cause damage to the carbonized fibers...”

Line 141: “Deconvolution arithmetic” → “Spectral deconvolution using five Raman bands...”

Line 159: “Time and thickness do not belong to a linear relationship” → “PyC thickness and deposition time show a nonlinear relationship...”

Response: Thank you for your valuable suggestions. We agree that these sentences and descriptions need to be improved. So, we have revised the sentences you mentioned and polished the language throughout the manuscript.

All the changes are highlighted in blue in Modified Version, locate at:

Abstract, lines 17-21;

Introduction. lines 35, 38-45, 59-61, 81, 82-85;

Experiment 2.2. lines 96-107;

Experiment 2.3. lines 113-118, 121-123;

Experiment 2.4. lines 126-132;

Results 3.1. lines 137-158, 166-172;

Results 3.2. lines 195-196, 224-231, 236-237, 240, 253-266, 274-283, 290-294, 306-322, 334-338, 342-353;

Conclusion, line 355, 356, 359-360, 363-365, 368-370.

Reviewer 3 Report

Comments and Suggestions for Authors

The main question addressed by this research is whether a uniform and stable Sc₂SnC MAX phase coating can be synthesized on SiC fibers using a molten salt method, and how this coating improves interface performance under irradiation and high-temperature conditions in SiCf/SiC composites. The study is both original and relevant within the field of advanced ceramic composites and nuclear materials, as it tackles a specific challenge: the structural and performance failure of traditional interface layers such as pyrolytic carbon and hexagonal boron nitride under neutron irradiation. These conventional materials tend to degrade, causing delamination and loss of mechanical integrity. The development of a MAX phase interface layer, particularly using Sc₂SnC, addresses the need for improved radiation and thermal stability. This work also fills a technical gap by demonstrating a novel method to uniformly coat SiC fibers with a MAX phase through molten salt synthesis—a method not previously applied successfully to Sc-containing phases in this context.

The conclusions presented in the study are well-supported by the experimental evidence. Characterization through SEM, XRD, and Raman spectroscopy confirms the successful formation and growth of the Sc₂SnC coating, while the morphological development from isolated flakes to a continuous layer aligns with observed reaction kinetics. The identification of ScCx as a key intermediate in the coating formation process adds depth to the mechanistic understanding and is logically consistent with the results. The study also honestly addresses the potential issues associated with the metastability of ScCx, including phase transformation and coating delamination, showing a thorough and balanced interpretation of the findings.

The references used throughout the paper are appropriate and relevant, citing foundational and recent work related to MAX phase materials, radiation stability, interface layer design, and coating techniques. These citations provide the necessary context to support the novelty and technical contribution of the work. Overall, the study demonstrates a meaningful advancement in the development of radiation- and heat-resistant interface coatings for SiC fiber-reinforced composites.

Technical notes:

I did not notice any typos, the paper is very well written. The text is professionally edited. The equations are clear, their format is nice.

However, there is a numbering issue:
In line 161 there is Equation (1). Hence, the Equations in lines 274-276 should not be (1)-(3), but (2)-(4). The referring text in line 258 should also be updated: (1-3) should be replaced by (2-4).

The figures are of good quality, suitable for publication.

There is a little problem though:
On Figure 1 the Raman spectra should be replaced by a better quality (resolution) images. They are slightly better in Figure 2, but these would also look better if their resolution is increased.

In conclusion: I recommend a minor revision, and when these issues are addressed, the paper can be accepted.

Author Response

Dear Reviewer:

Thank you for your comments on improving the quality of our manuscript. We have made the necessary modifications based on your feedback, which have been marked in the “Modified Version for Comparison - Materials-3610652”. (Complete version for reading is also provided.) The details of the changes are as follows:

Reviewer #3: The main question addressed by this research is whether a uniform and stable Sc2SnC MAX phase coating can be synthesized on SiC fibers using a molten salt method, and how this coating improves interface performance under irradiation and high-temperature conditions in SiCf/SiC composites. The study is both original and relevant within the field of advanced ceramic composites and nuclear materials, as it tackles a specific challenge: the structural and performance failure of traditional interface layers such as pyrolytic carbon and hexagonal boron nitride under neutron irradiation. These conventional materials tend to degrade, causing delamination and loss of mechanical integrity. The development of a MAX phase interface layer, particularly using Sc2SnC, addresses the need for improved radiation and thermal stability. This work also fills a technical gap by demonstrating a novel method to uniformly coat SiC fibers with a MAX phase through molten salt synthesis—a method not previously applied successfully to Sc-containing phases in this context.

The conclusions presented in the study are well-supported by the experimental evidence. Characterization through SEM, XRD, and Raman spectroscopy confirms the successful formation and growth of the Sc2SnC coating, while the morphological development from isolated flakes to a continuous layer aligns with observed reaction kinetics. The identification of ScCx as a key intermediate in the coating formation process adds depth to the mechanistic understanding and is logically consistent with the results. The study also honestly addresses the potential issues associated with the metastability of ScCx, including phase transformation and coating delamination, showing a thorough and balanced interpretation of the findings.

The references used throughout the paper are appropriate and relevant, citing foundational and recent work related to MAX phase materials, radiation stability, interface layer design, and coating techniques. These citations provide the necessary context to support the novelty and technical contribution of the work. Overall, the study demonstrates a meaningful advancement in the development of radiation- and heat-resistant interface coatings for SiC fiber-reinforced composites.

Technical notes:

I did not notice any typos; the paper is very well written. The text is professionally edited. The equations are clear; their format is nice.

Response: Really appreciate your time on our manuscript. According to your valuable comments, we have made corresponding modification and polishing to the manuscript. Following are the point-to-point responses.

1). However, there is a numbering issue:

In line 161 there is Equation (1). Hence, the Equations in lines 274-276 should not be (1)-(3), but (2)-(4). The referring text in line 258 should also be updated: (1-3) should be replaced by (2-4).

The figures are of good quality, suitable for publication.

Response: Appreciate your time and thank you for the valuable suggestions. The Equation number may cause misunderstanding and should be revised.

Changes are marked in lines 273, 287-289, highlighted in red.

2). There is a little problem though:

On Figure 1 the Raman spectra should be replaced by a better quality (resolution) images. They are slightly better in Figure 2, but these would also look better if their resolution is increased.

Response: Thank you for your valuable comment. For a better reading experience, we have re-uploaded the figures with highest resolution.

Changes in all Figures.

Round 2

Reviewer 1 Report

Comments and Suggestions for Authors

The authors have performed a lot of improvements. Now the paper can be published.